# Design and Experimental Research of 3-RRS Parallel Ankle Rehabilitation Robot

**DOI:** 10.3390/mi13060950

**Published:** 2022-06-16

**Authors:** Yupeng Zou, Andong Zhang, Qiang Zhang, Baolong Zhang, Xiangshu Wu, Tao Qin

**Affiliations:** 1College of Mechanical and Electronic Engineering, China University of Petroleum (East China), Qingdao 266580, China; zouyupeng@upc.edu.cn (Y.Z.); s20040049@s.upc.edu.cn (A.Z.); s19040017@s.upc.edu.cn (Q.Z.); s19040027@upc.edu.cn (B.Z.); s20040023@s.upc.edu.cn (X.W.); 2Xiangyang Key Laboratory of Rehabilitation Medicine and Rehabilitation Engineering Technology, Hubei University of Arts and Science, Xiangyang 441053, China; 3School of Mechanical Engineering, Hubei University of Arts and Science, Xiangyang 441053, China

**Keywords:** ankle rehabilitation, parallel mechanism, simulation analysis, prototype experiment

## Abstract

The ankle is a crucial joint that supports the human body weight. An ankle sprain will adversely affect the patient’s daily life, so it is of great significance to ensure its strength. To help patients with ankle dysfunction to carry out effective rehabilitation training, the bone structure and motion mechanism of the ankle were analyzed in this paper. Referring to the configuration of the lower-mobility parallel mechanism, a 3-RRS (R and S denote revolute and spherical joint respectively) parallel ankle rehabilitation robot (PARR) was proposed. The robot can realize both single and compound ankle rehabilitation training. The structure of the robot was introduced, and the kinematics model was established. The freedom of movement of the robot was analyzed using the screw theory, and the robot kinematics were analyzed using spherical analytics theory. A circular composite rehabilitation trajectory was planned, and the accuracy of the kinematics model was verified by virtual prototype simulation. The Multibody simulation results show that the trajectory of the target point is basically the same as the expected trajectory. The maximum trajectory error is about 2.5 mm in the simulation process, which is within the controllable range. The experimental results of the virtual prototype simulation show that the maximum angular deflection error of the three motors is 2° when running a circular trajectory, which meets the experimental requirements. Finally, a control strategy for passive rehabilitation training was designed, and the effectiveness of this control strategy was verified by a prototype experiment.

## 1. Introduction

Ankle sprains are a common type of musculoskeletal system injury, accounting for about 7% to 10% of patients with sports injuries [1]. When the human body performs upright sporting activities, the body weight is loaded on the ankle joint on both sides. Especially for athletes, during strenuous exercise, the load and impact on the ankle rise sharply, which easily leads to ankle joint overload and sprains [2,3]. If patients with an ankle sprain injury undertake rehabilitation training too late or inefficiently, it is easy for this injury to escalate to more serious conditions, such as chronic ankle instability [4]. Medical theory and clinical experiments proved that effective rehabilitation training can accelerate the process of ankle rehabilitation; the earlier the rehabilitation training is undertaken, the more conducive it is to the recovery of motor function [5]. However, the number of patients with ankle sprains is large, and the number of doctors who specialize in rehabilitation training is seriously inadequate [6]. Traditional manual rehabilitation training has low efficiency, and there are no uniform training parameters and indicators [7]. Therefore, it is of great relevance to develop an ankle rehabilitation robot to help patients with ankle joint dysfunction.

Ankle rehabilitation robots have been a research hotspot in medical rehabilitation in the past few decades. According to the interactive human–machine spectrum, ankle rehabilitation robots can be divided into the wearable type and the pedal type [8,9]. Anklebot, a wearable ankle rehabilitation robot proposed by MIT, was powered by two parallel linear motors. Anklebot can realize the passive training of two degrees of freedom, according to the predetermined trajectory [10,11]. Based on the Stewart platform, the Tokyo University of Science and Technology designed an ankle rehabilitation exoskeleton. The robot was driven by six linear cylinders. The pedal ankle rehabilitation robot holds the foot in position on the pedal, and the pedal drives the ankle to rotate for rehabilitation training [12,13,14]. Based on the Agile Eye structure, the National Research Council designed an ankle rehabilitation robot called PKAnkle, which can effectively avoid joint dislocation during the rehabilitation process. What is more, PKAnkle can detect the interaction force and torque between patient and robot using a pressure sensor [15]. Wiggin et al. of Carnegie Mellon University designed a spring-loaded energy-storage ankle rehabilitation device. The spring was used as an energy storage device to assist in rehabilitation by analyzing the posture of the ankle joint during movement [16]. Jamwal presented the first ankle rehabilitation exoskeleton powered by PMA. The robot uses a fuzzy logic controller to achieve the required degrees of freedom for the robot. A web-based communication protocol between the patient and the medical practitioner is also proposed. The importance of human–robot interaction is also discussed elsewhere [17,18]. Park et al. designed a flexible active ankle rehabilitation exoskeleton by simulating the muscle-tendon-ligament-skin structure. The robot was able to perform controlled ankle rehabilitation movements in different sagittal planes [19,20,21]. Latifah et al. designed a reconfigurable ankle rehabilitation device, based on a 3-RPS parallel mechanism. MICD (maximum internal connecting circle radius) performance metrics were analyzed for different motion states. Structural optimization was performed to ensure the output [22]. Florida International University has designed an ankle rehabilitation device based on pneumatic artificial muscle actuation. The device collects electromyographic signals from the patient’s lower leg to regulate the power required for rehabilitation and to assist the patient’s rehabilitation [23,24]. Jungwon Yoon et al. designed a reconfigurable ankle rehabilitation robot with multiple rehabilitation modes. The robot achieves three-degrees-of-freedom rehabilitation movements of the ankle joint while completing rehabilitation movements of the toes [25]. Based on a lower-mobility parallel mechanism, the University of Auckland developed a pedal ankle rehabilitation robot. The robot can realize three rotational degrees of freedom of the ankle joint, with high control accuracy [26].

In conclusion, many institutions have conducted research into ankle rehabilitation robots and have achieved diverse results. The ankle rehabilitation robot has many existing problems, such as a complex structural design, insufficient or redundant degrees of freedom, and non-existent rotational centering. Therefore, improvements should be made in the aspects of structural design and human-machine compatibility.

In terms of lower-mobility parallel mechanisms, a parallel ankle rehabilitation robot (PARR) is proposed in this paper. Essentially, PARR is a 3-RRS parallel mechanism. This mechanism can rotate around the rotation center to match the ankle motion mechanism. As a parallel mechanism, PARR has the characteristics of a typical parallel mechanism. In addition, PARR offers other properties, such as a simple and compact structure and high human–machine compatibility, so it is suitable for ankle rehabilitation training.

The rest of this article is organized as follows. In the second part, the bone structure and motion mechanism of the ankle are analyzed, and the mechanical structure of the PARR is introduced. In the third part, the kinematics model and the kinematic analysis are explained. In the fourth part, we show a simulation model that was built using Multibody. Kinematic simulation analysis was conducted to verify the safety and feasibility of the PARR. The passive rehabilitation training control strategy was designed and the prototype experiment was carried out to compare and analyze the feasibility of the robot to achieve rehabilitation training. Finally, our conclusions are presented and further work is forecasted.

## 2. Mechanical Structure

### 2.1. Ankle Motion Mechanism

The ankle joint is the most heavily loaded joint in the human body and is involved in the majority of movement postures of the lower limbs. Its bone structure consists of the tibia, the fibula’s distal end, and the trochlea talus (see Figure 1a). The coordination of the ankle joint and knee joint enables the foot to adapt to walking on various surfaces [27,28].

The ankle joint has three motion forms: dorsiflexion/plantar flexion, adduction/abduction, and varus/valgus. The instep and the shank form a right angle when they are in a normal state. When the toe moves downward, the angle between the shank and the instep increases gradually, which is plantar flexion. The opposite movement is dorsiflexion. The outward rotation of the foot around the shank is known as abduction, and the opposite movement is known as adduction. The movement of raising the internal margin of the foot and lowering the external margin is varus, and the opposite movement is valgus. The motion model is constructed based on the form of ankle motion, as shown in Figure 1b.

The motion range of the ankle joint is affected by many factors, such as the joint surface area, the strength of the ligaments, and the volume and elasticity of muscles [29,30]. The motion range of the ankle is shown in Table 1. The talus is wide in the front and narrow in the back. When the ankle is in dorsiflexion, the wider part of the talus enters the malleolar cave. This can prevent excessive movement of the talus. The ankle joint is in a stable fastening state. When the ankle is in plantar flexion, the narrow part of the talus enters the malleolar cave. The ankle joint is in a loose state, and the talus can move in offset to the two sides. It is easy to cause a dislocation sprain of the ankle at this time. Compared to the medial malleolus, the lateral malleolus is long and low. Therefore, it is easy to cause an ankle varus injury in sports.

Rehabilitation training after ankle damage can improve the strength of the muscles related to ankle movement, stimulate the activation of related neural mechanisms, and promote the recovery of normal ankle function. To achieve comprehensive ankle rehabilitation training, the ankle rehabilitation robot needs to be designed to work in two modes: single-DOF (degree of freedom) rehabilitation training with only one form of exercise, and multi-DOF compound rehabilitation training with two or more forms of exercise simultaneously.

### 2.2. Mechanical Structure of the 3-RRS PARR

Compared to the 6-DOF parallel mechanism, the lower-mobility parallel mechanism has a concise configuration and a simple control strategy. This lower-mobility parallel mechanism is acceptable for ankle rehabilitation training [31,32]. Based on the 3-RRS parallel mechanism, a novel parallel ankle rehabilitation robot was proposed.

The 3-RRS PARR has three degrees of rotational freedom, and its rotation center is fixed in the space. Through the design of the robot structure, the rotation center can coincide with the patient’s ankle joint during the rehabilitation training process. Except for the rotation of three degrees of freedom, the ankle does not have axial movement during rehabilitation. The 3D model of the 3-RRS PARR was shown in Figure 2.

The static platform and the moving platform of the 3-RRS PARR are connected by three branch chains. Each branch chain has three kinematic pairs. The branch chain is an RRS configuration, and kinematic pairs of the branch chains are the same. The ends of the branch chains are connected to the moving platform through a spherical joint. The rotation axes of the base shafts of three branch chains coincide on a common axis. The rotation axes of middle revolute pairs are perpendicular to each other in space. The common axis of the branch chains at the base revolute pair and the middle rotation axis intersect at a fixed point in space. The moving platform has a 3-DOF to rotate around the fixed point. The PARR is driven by a low-speed torque motor. Compared with a high-speed DC motor, a low-speed torque motor can work continuously in a locked-rotor state. Besides the motor, an adjustable pedal is installed on the moving platform. By adjusting the height and the length of the adjustable pedal, the patient’s ankle joint coincides with the rotation center. This structural design ensures the patient’s safety in the process of rehabilitation training, avoids secondary injury, and realizes human–machine compatibility.

The composition of the RRS branch chain is shown in Figure 3. The incremental encoder is installed on the motor, and the movement data of the crankshafts are monitored through the encoder. Both the motor base and crankshaft are installed on the static platform. The crankshaft and the static platform are connected through the base revolute pair (R). The crankshaft and the linkage are connected through the middle revolute pair (R). Finally, the linkage is connected to the moving platform through the spherical joint (S).

## 3. Robot Kinematics Analysis

### 3.1. Robot Kinematics Modeling

According to the D-H matrix, the kinematics model of the 3-RRS parallel mechanism is established (see Figure 4). The direction vectors connecting the center of the kinematic pair and the rotation center are expressed as *u_i_*, *v_i_*, *w_i_* (i = 1, 2, 3). The static coordinate system is *O*-*XYZ*, the origin of which is the rotation center *O* of the mechanism. The *z*-axis is perpendicular to the static platform, and the *x*-axis is in the plane formed by the vector w1 and the *z*-axis. The *y*-axis is determined according to the *x*-axis and *z*-axis with the right-handed spiral rule. The moving coordinate system is *O*-*xyz*, the origin of coordinates is rotation center *O*. The axes of *Ox*, *Oy*, *Oz* are respectively along the direction of *w_i_*. The angles between the axis of the middle revolute pair and the vertical direction in the moving coordinate system are both 54.74°. Therefore, the axes of the direction vectors *w_i_* and *v_i_* are perpendicular to each other in space and intersect at rotation point *O*.

### 3.2. DOF Analysis of PARR

The DOF of the 3-RRS parallel mechanism is analyzed using the screw theory, which determines the common and virtual constraints of the mechanism [33]. The twist system of the branch chain can be expressed as:(1)$$\left\{ \begin{array}{l} {＄}_{i1}=(u_{i};0)\\ {＄}_{i2}=(v_{i};0)\\ {＄}_{i3}=(w_{i};0) \end{array}\right.,$$

The three kinematic pairs of each branch chain have different axes in space and intersect at point *O*. Therefore, *$**_i_*_1_, *$**_i_*_2_, *$**_i_*_3_ is linearly independent. According to the twist system, the basic screw system of the reverse screw can be obtained as:(2)$$\left\{ \begin{array}{l} {＄}_{1}{}^{T}=(1\;0\;0;\;0\;0\;0)\\ {＄}_{2}{}^{T}=(0\;1\;0;\;0\;0\;0)\\ {＄}_{3}{}^{T}=(0\;0\;1;\;0\;0\;0) \end{array}\right.$$

According to the relationship between the wrench and twist in the screw theory, it can be established from the reverse screw that the branch chain is subject to forces in the directions of the *x*-axis, *y*-axis, and *z*-axis, without the force couple. Therefore, each branch chain has three rotational DOF, and there is no moving DOF. There is a spherical joint in the branch chain. Therefore, *$**_i_*_3_ = (*w_i_*; 0) = (*a*_1_, *a*_2_, *a*_3_; 0 0 0), where *a*_1_, *a*_2_, *a*_3_ is not 0. The twist of the spherical joint can be converted into:(3)$${＄}_{i3}=\left\{ \begin{array}{l} (a_{1}\;0\;0;\;0\;0\;0)\\ (0\;a_{2}\;0;\;0\;0\;0)\\ (0\;0\;a_{3};\;0\;0\;0) \end{array}\right..$$

Due to the existence of the spherical joint, according to the analysis of the branch chain’s DOF and the spherical joint, it can be found that the branch chain has two local DOF. Therefore, the motion of the branch chain can be simplified to the motion of the 3R structure. Since the whole mechanism consists of three branch chains, the general wrench consists of three branches of the wrench. The wrench of the 3-RRS parallel mechanism can be reduced to a base wrench system.
(4)$$＄=\left\{ \begin{array}{l} (1\;0\;0;\;0\;0\;0)\\ \left(0\;1\;0;\;0\;0\;0\right)\\ \left(0\;0\;1;\;0\;0\;0\right) \end{array}\right.$$

Thus, the general constraint of the mechanism is 3. According to Equation (4), it is established that the mechanism is subject to forces in the direction of the *x*-axis, *y*-axis, and *z*-axis, and is not affected by the force couple. The inverse wrench of Equation (4) is deduced as:(5)$${＄}^{T}=\left\{ \begin{array}{l} (1\;0\;0;\;0\;0\;0)\\ \left(0\;1\;0;\;0\;0\;0\right)\\ \left(0\;0\;1;\;0\;0\;0\right) \end{array}\right.$$

Equation (5) establishes the twist corresponding to the general wrench. It can be seen that the mechanism has three mutually perpendicular rotational degrees of freedom in space [34].

According to the modified Kutzbach–Gruble equation, the DOF of the mechanism can be calculated as:(6)$$M=d(n-g-1)+\sum_{i=1}^{g}f_{i}+v-\varsigma =3$$

According to Equation (6), M is the number of degrees of freedom of the mechanism; d is the number of orders of the mechanism; n is the number of members in the mechanism; g is the number of kinematic pairs in the mechanism; $f_{i}$ is the number of degrees of freedom of the i-th kinematic sub; λ is the number of general constraints of the mechanism; and ς is the number of local degrees of freedom of the mechanism.

According to the screw theory analysis results, the number of DOF corresponds to the rotation around the three axes. There are no redundant DOF in other directions. That is to say, the rotation center of the PARR is consistent with that of the ankle joint, which meets the requirements of ankle rehabilitation.

### 3.3. Inverse Kinematics Model

Here, *α, β,* and *γ* represent the attitude angles of the moving platform rotating around the static coordinate system’s *z*-axis, *y*-axis, and *x*-axis, respectively. The process of solving the three input angles *θ_i_* (*i* = 1, 2, 3) of the mechanism through the attitude change angle *α*, *β*, *γ* of the moving platform is by the inverse kinematics solution of the PARR [35]. When the moving platform and the static platform are parallel, the position of the mechanism is defined as the initial pose. The angle between *u_i_* and *v_i_* is δ1, the angle between *v_i_* and *w_i_* is *δ*_2_, the angle between *w_i_* and the *z*-axis of the static coordinate system is *ε*_1_, and the angle between plane *YOZ* and plane *w_i_OZ* is *ε_i_*.

When the PARR is in the initial pose, the direction vector connecting the rotation center of the mechanism and the center of the spherical joint can be expressed as:(7)wi0=(sinεisinϕ1cosεisinϕ1 cosϕ1)T

When the pose of the moving platform changes, the direction vector can be expressed as:(8)$$w_{i}=Rw_{i0}$$

The Euler rotation matrix is *R* = *R*(*z*,*a*)*R*(*y*,*b*)*R*(*x*,*y*).

According to the spherical analytic theory, the direction’s cosine of *v*_1_ in the initial pose can be expressed as:(9)$$v_{10}=\left[\sin\delta_{1}\sin\theta_{1}\sin\delta_{1}\cos\theta_{1}\cos\delta_{1}\right]$$

According to the structural features of the PARR, the vectors of the middle revolute pairs of each branch chain are distributed uniformly; then, *v*_2_ and *v*_3_ can be obtained by rotating *v*_1_ in the negative direction about the *Z*-axis, where *ε_ij_* is the angle between plane *w_i_OZ* and plane *w_j_OZ*. The general expression of the cosine of the vector of the middle revolute pair direction is as follows:(10)$$v_{i}=R_{\varepsilon_{i}}v_{1}=\left[\begin{array}{c} \cos\varepsilon_{i}\sin\delta_{1}\sin\theta_{i}-\sin\varepsilon_{i}\sin\delta_{1}\cos\theta_{i}\\ \sin\varepsilon_{i}\sin\delta_{1}\sin\theta_{i}-\cos\varepsilon_{i}\sin\delta_{1}\cos\theta_{i}\\ \cos\delta_{1} \end{array}\right]$$
$$i=1,2,3(\varepsilon_{i}=0,\frac{2}{3}\pi ,\frac{4}{3}\pi )$$

According to the structural feature of the linkage, the two vectors *v_i_* and *w_i_* form a fixed angle, and the constraint equation is constructed based on this condition:(11)$$v_{i}{}^{T}\cdot w_{i}=\cos\delta_{2}$$

Substituting and simplifying the constraint equation can be obtained with:(12)$$A_{i}\sin\theta_{i}+B_{i}\cos\theta_{i}=C_{i}$$

*A_i_*, *B_i_*, and *C_i_* are expressions containing structural parameters, input angles, and output angles, assuming $x_{i}=\tan\frac{\theta_{i}}{2}$, then $\sin\theta_{i}=\frac{2x_{i}}{1+x_{i}^{2}},\cos\theta_{i}=\frac{1-x_{i}^{2}}{1+x_{i}^{2}}$, by trigonometric substitution. Equation (12) can be expressed as:(13)$$\left(B_{i}+C_{i}\right)x_{i}^{2}-2A_{i}x_{i}+\left(C_{i}-B_{i}\right)=0$$

When the attitude angle of the moving platform is known, input angles of the branch chains can be obtained:(14)$$\theta_{i}=2\arctan x_{i}$$

### 3.4. Forward Kinematics Model

The process of solving the attitude change angles *α*, *β*, and *γ* of the moving platform through three input angles via *θ_i_* (*i* = 1, 2, 3) is the forward kinematics solution of the PARR. The angle between plane *v*_1_*Ow*_1_ and plane *u*_1_*Ov*_1_ is *ψ*_1_, and the angle between plane *v*_1_*Ow*_1_ and plane *w*_1_*Ow*_2_ is *ψ*_2_. According to the spherical analytic theory, the motion chain of the spherical polygon (see Figure 5) is established [36,37].

The cosine of the direction vector of *w*_1_, *w*_2_, and *w*_3_ is expressed by spherical polygons 4321, 54,321, and 5’4321, respectively. In addition, *v_i_* has been solved in the inverse kinematics solution. Substituting *w_i_* and *v_i_* into the constraint equation of Equation (11) can be simplified as follows:(15)$$D_{i}\sin\psi_{1}+E_{i}\cos\psi_{1}+F_{i}=0$$

*D_i_*, *E_i_*, and *F_i_* are expressions containing the structural parameters, input angle, external angle, and parameters *ξ*_1_ and *ξ*_2_. The unknown quantity *ψ_i_* can be solved using a trigonometric function, as follows:(16)$$\sin\psi_{1}=\frac{E_{1}F_{2}-E_{2}F_{1}}{D_{1}E_{2}-D_{2}E_{1}}\;\cos\psi_{1}=\frac{D_{1}F_{2}-D_{2}F_{1}}{D_{2}E_{1}-D_{1}E_{2}}$$

When *D*_1_*E*_2_ − *D*_2_*E*_1_≠ 0:(17)$$E_{1}^{2}F_{2}^{2}+E_{2}^{2}F_{1}^{2}+D_{1}^{2}F_{2}^{2}+D_{2}^{2}F_{1}^{2}-D_{1}^{2}E_{2}^{2}-D_{2}^{2}E_{1}^{2}\;-2E_{1}E_{2}F_{1}F_{2}-2D_{1}D_{2}F_{1}F_{2}+2D_{1}D_{2}E_{1}E_{2}=0$$

In order to solve this efficiently, assuming $y=\tan\frac{\psi_{2}}{2}$, then $\sin\theta_{i}=\frac{2\psi_{i}}{1+\psi_{i}^{2}}$, $\cos\theta_{i}=\frac{1-\psi_{i}^{2}}{1+\psi_{i}^{2}}$. Equation (17) can be reorganized as follows:(18)$$\overset{8}{\underset{i=0}{\sum }}N_{i}y^{i}=0$$

Analyzing Equation (18) shows that there are eight sets of solutions for y. According to the structural features of the PARR, it can be seen that there is a unique solution for the mechanism, by substituting the solved *ψ*_1_ and *ψ*_2_ into *w_i_*. According to Equation (8), the following equation can be obtained:(19)$$R=w_{i}w_{i0}{}^{-1}=\left[\begin{array}{c} n_{11}\;n_{12}\;n_{13}\\ n_{21}\;n_{22}\;n_{23}\\ n_{31}\;n_{32}\;n_{33} \end{array}\right]$$

According to the matrix element correspondence principle, the corresponding attitude angle of the moving platform is:(20)$$\begin{array}{l} \alpha =\arcsin(\frac{n_{23}}{\cos\beta })\\ \beta =\arcsin(-n_{31})\\ \gamma =\arccos(\frac{n_{11}}{\cos\beta }) \end{array}$$

### 3.5. Numerical Example

Setting the mechanical structure angles *δ*_1_ at 54.74°, and the mechanical structure angles *δ*_2_ at 40°, based on the inverse kinematics model, the relationship between *θ_i_* (*i* = 1, 2, 3) and *α*, *β*, and *γ* are established, as shown in Figure 6. When the PARR performs varus/valgus rehabilitation training at the range of −30°~30°, the relationship between *θ_i_* (*i* = 1, 2, 3) and *γ* is as shown in Figure 6a. When the PARR performs dorsiflexion/plantar flexion rehabilitation training in the range of −40°~30°, the relationship between *θ_i_* (*i* = 1, 2, 3) and *β* is as shown in Figure 6b. When the PARR performs adduction/abduction rehabilitation training in the range of −30°~30°, the relationship between *θ_i_* (*i* = 1, 2, 3) and *α* is as shown in Figure 6c.

There is no obvious mutation in the curve. It is evident that if stabilized input is given to the motor, the PARR can move smoothly. Compared with the motion range of the ankle given in Table 1, it can be seen that the rehabilitation training of all motions can be completed at 100%. Therefore, it was shown that the PARR system meets the actual rehabilitation training requirements.

## 4. Multibody Simulation Analysis

### 4.1. Simulation Model

To verify the accuracy of the kinematics model and the feasibility of achieving ankle rehabilitation training, the simulation model of the PARR was established with the MATLAB Simulink toolbox (see Figure 7a).

Signal alpha, signal beta, and signal gamma represent the rotational angle of the moving platform around the static coordinate system *O*-*XYZ*. The inverse kinematics model module can calculate the motion law of each motor. The PARR model module was used for the PARR simulation model. The transform sensor module is used to extract simulation data. The kinematics simulation results are compared and analyzed with the theoretical values. The picture of the PARR module is shown in Figure 7b. The picture of the Branch Simulation is shown in Figure 7c. It consists of a joint and mechanism, where the joint is two revolute joints and a spherical joint, and the mechanism is a crankshaft, linkage, and moving platform. The simulation model of the PARR is shown in Figure 8.

### 4.2. Kinematics Simulation Analysis

The feasibility of single-DOF rehabilitation training is shown in Figure 6. To realize the requirements of multi-DOF compound rehabilitation training, a circular motion trajectory with a radius of 87.5 mm on the *YOZ* projection plane was planned for this paper. This trajectory can realize compound rehabilitation training of ankle dorsiflexion/plantarflexion and adduction/abduction. The target point is 200 mm along the positive direction of the *OX* coordinate axis, as shown in Figure 9a. The movement state of the PARR simulation model under this trajectory is shown in Figure 9b.

The motion trajectory of the target point projected on the *YOZ* surface is shown in Figure 10a. The error curve of the trajectory is shown in Figure 10b. The simulation results show that the trajectory of the target point is basically the same as the expected trajectory. The maximum trajectory error is about 2.5 mm in the simulation process; the smaller errors may be due to model errors in the modeling process, which are within the controllable range. The curve has no large mutation. During ankle rehabilitation training, the gravitational and inertial forces on the patient’s leg are mainly carried by the support mechanism, such as the seat. Therefore, the external forces applied to the moving platform are small. For the robot driving torque simulation analysis, a vertical downward force of 10 N is applied at the rotational centering point to simulate the external force applied during the rehabilitation movement. The driving torque is monitored by the simulation module. The curve of the motor driving torque is shown in Figure 10c. T1, T2, and T3 represent the motor driving torque. The smooth driving torque change curve is shown without sudden change. The prototype is driven by a 4.4 N·m DC torque motor, to meet the driving torque requirements.

This verifies the feasibility of the PARR model designed in this paper for rehabilitation training.

## 5. Prototype Experiment

In the early stage of the rehabilitation process, the passive rehabilitation training mode is mainly used to train the range of motion of the patient’s ankle joint. Since the ankle joint is relatively stiff during this period, the patient cannot carry out activities by him- or herself; therefore, passive rehabilitation training should be carried out with the help of external forces. Passive rehabilitation training requires the following error to be as small as possible, so the position-speed double closed-loop control strategy was adopted, as shown in Figure 11. The inner loop is the speed control loop, using a proportional-integral controller, and the outer loop is the position control loop, using a proportional controller.

The PARR experimental prototype was designed to test the circular rehabilitation motion track planned above. The experimental device is shown in Figure 12a. The experimental prototype is shown in Figure 12b. The encoder was used to measure the motor rotation. The adjustable pedal was removed to monitor the corner of the moving platform by Angle Sensor 1. Angle Sensor 2 was used for leveling the moving platform.

The circular trajectory motion period was set to 7 s, and the rehabilitation trajectory motion experiment was carried out. As shown in Figure 13, it was verified that the PARR can drive the affected limb to complete the rehabilitation movement represented in the experiment, simulating the compound rehabilitation movement of an ankle joint. It can be seen from Figure 14a–d that the actual rotation law of the three motors can follow the expected motion law. The maximum following error is about 2°, which meets the experimental requirements. It can be seen from Figure 14e,f that the rotation law of the rotating center of the prototype test experiment conforms to expectations, without substantial mutation, and with a small error. The feasibility of the ankle rehabilitation robot was verified. During the experimental process, the small error in the turning angle of the dynamic platform may have been caused by an installation error in the prototype. In the subsequent experimental research, the robot prototype structure should be improved to eliminate and avoid the influence of prototype installation errors on the experimental results. In addition, dynamics analysis should be added to plan the output force of the resulting dynamic platform.

## 6. Conclusions

To help patients with ankle dysfunction in rehabilitation training, based on the lower-mobility parallel mechanism, a novel 3-RRS parallel ankle rehabilitation robot was proposed in this paper. To obtain better kinematic characteristics in the process of rehabilitation training, the bone structure and motion mechanism of the ankle were analyzed, and the motion range of the ankle joint was detailed. Then, the structure and the kinematics model of the PARR were introduced. The DOF of the PARR was calculated using the screw theory, and the kinematics analysis of the PARR was analyzed with the spherical analytic theory. The feasibility of single-DOF rehabilitation training was proved with relevant numerical examples. The PARR simulation model was established by the MATLAB Simulink toolbox. A circular trajectory was planned and simulated. The Multibody simulation results show that the trajectory of the target point is basically the same as the expected trajectory. The maximum trajectory error is about 2.5 mm in the simulation process, which is within the controllable range. The feasibility of multi-DOF rehabilitation training was verified through a simulation analysis of the circular trajectory. A passive rehabilitation training control strategy was designed according to the rehabilitation training requirements. The experimental results of the virtual prototype simulation show that the maximum angular deflection error of the three motors is 2° when running a circular trajectory, which meets the experimental requirements. Based on the above research, the feasibility of the 3-RRS PARR proposed in this paper was proved, and the foundation for the human-machine experiments was laid.

Regarding the existing research on ankle rehabilitation robots, comfort during rehabilitation is one of the urgent issues to be addressed. In follow-up research, the structural improvement and kinetic analysis of PARR should be combined with rehabilitation medicine to further improve human–machine compatibility. Further human–machine experiments will be conducted in the future to evaluate the reliability of the robot system.

## Figures and Tables

**Figure 1 micromachines-13-00950-f001:**
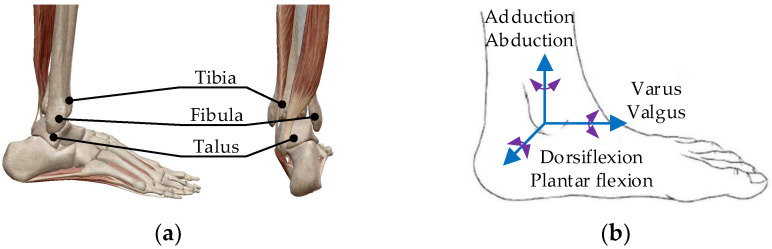
(**a**) The ankle bone structure; (**b**) the ankle motion model.

**Figure 2 micromachines-13-00950-f002:**
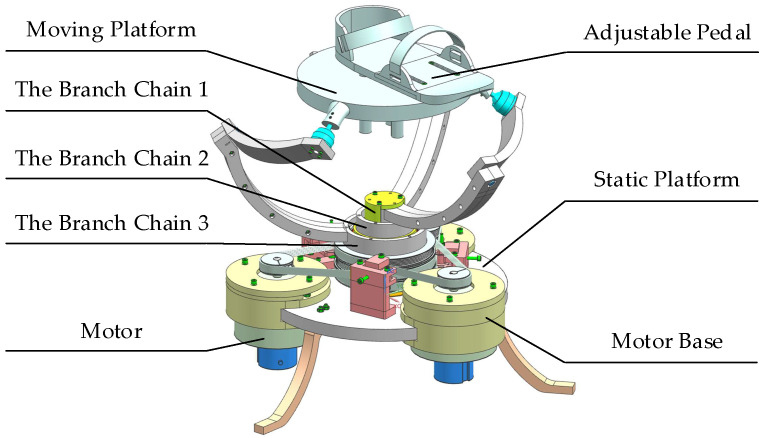
Branch-chain structure of RRS.

**Figure 3 micromachines-13-00950-f003:**
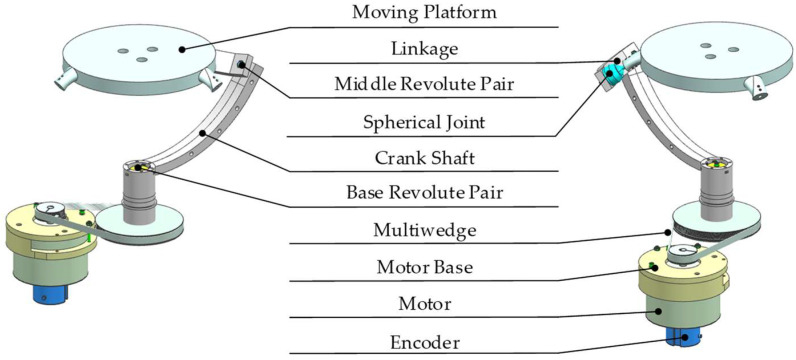
Branch-chain structure of RRS.

**Figure 4 micromachines-13-00950-f004:**
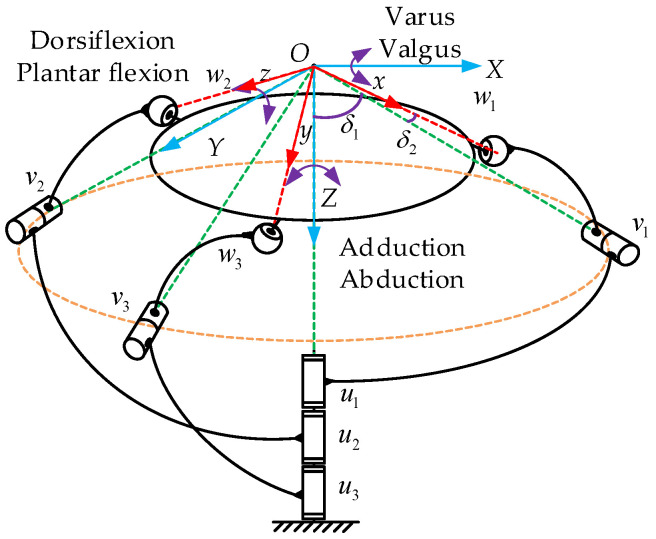
Kinematics model of the 3-RRS parallel mechanism.

**Figure 5 micromachines-13-00950-f005:**
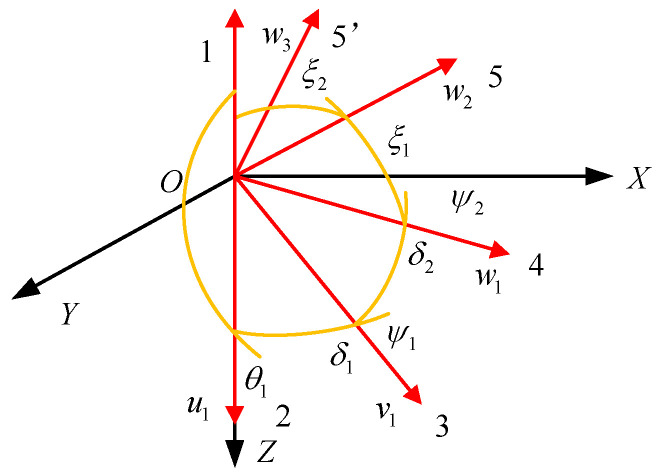
Motion chain of the spherical polygon.

**Figure 6 micromachines-13-00950-f006:**
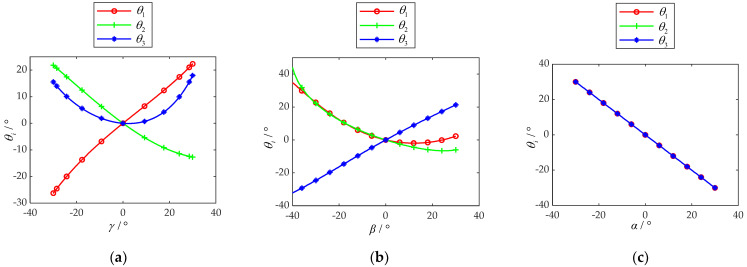
(**a**) Relationship between *θ_i_* and *γ*, when *α* = 0°, *β* = 0°; (**b**) relationship between *θ_i_* and *β,* when *α* = 0°, *γ* = 0°; (**c**) relationship between *θ_i_* and *α,* when *β* = 0°, *γ* = 0°.

**Figure 7 micromachines-13-00950-f007:**
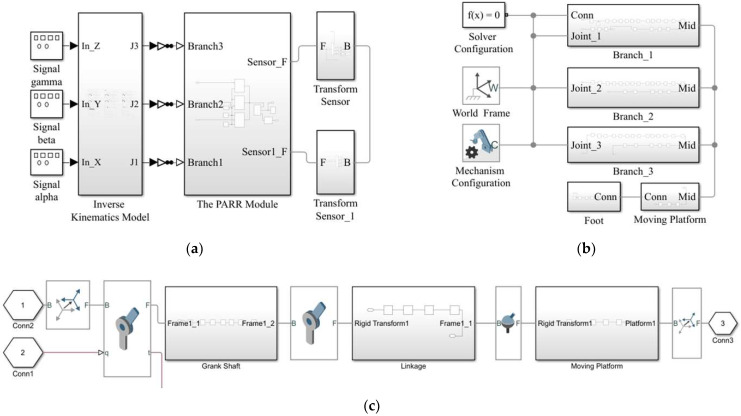
(**a**) Simulation module construction; (**b**) the PARR model; (**c**) the branch simulation.

**Figure 8 micromachines-13-00950-f008:**
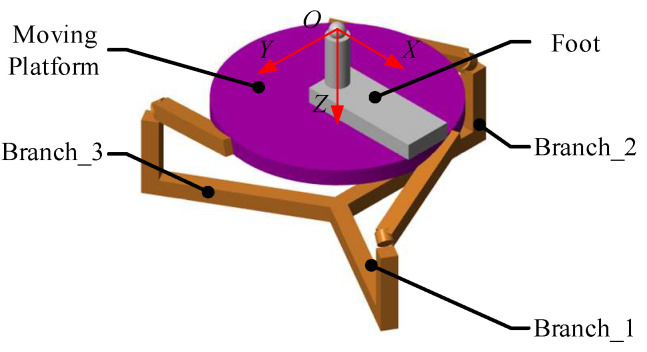
Simulation model of the PARR system.

**Figure 9 micromachines-13-00950-f009:**
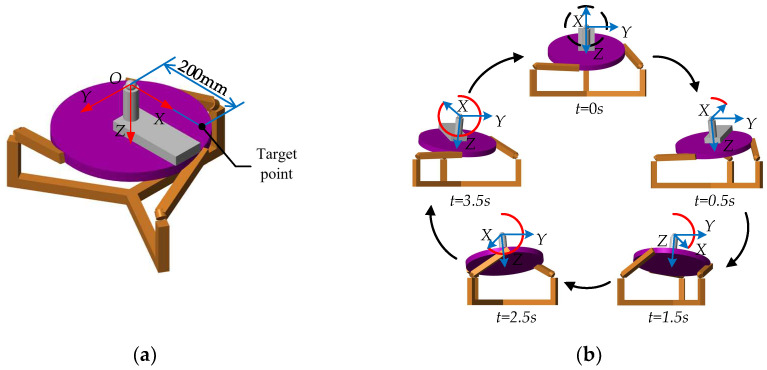
(**a**) Target point; (**b**) circular trajectory.

**Figure 10 micromachines-13-00950-f010:**
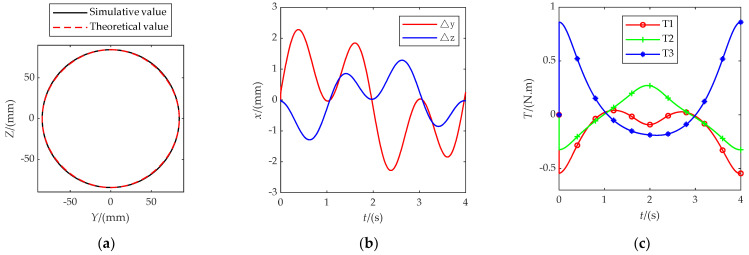
(**a**) Simulation result of circular trajectory; (**b**) the error curve of the trajectory; (**c**) the curve of the motor-driving torque.

**Figure 11 micromachines-13-00950-f011:**
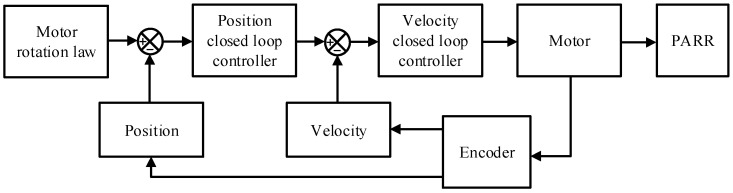
The control strategy.

**Figure 12 micromachines-13-00950-f012:**
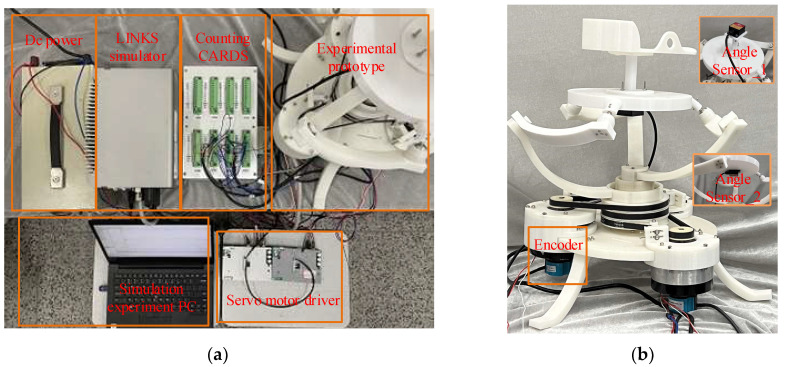
(**a**) The experimental device. (**b**) The error curve of the trajectory.

**Figure 13 micromachines-13-00950-f013:**
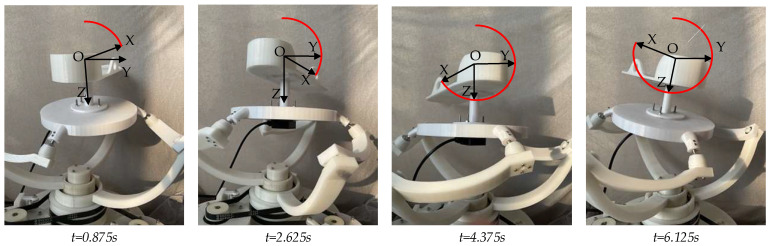
Prototype experiment track.

**Figure 14 micromachines-13-00950-f014:**
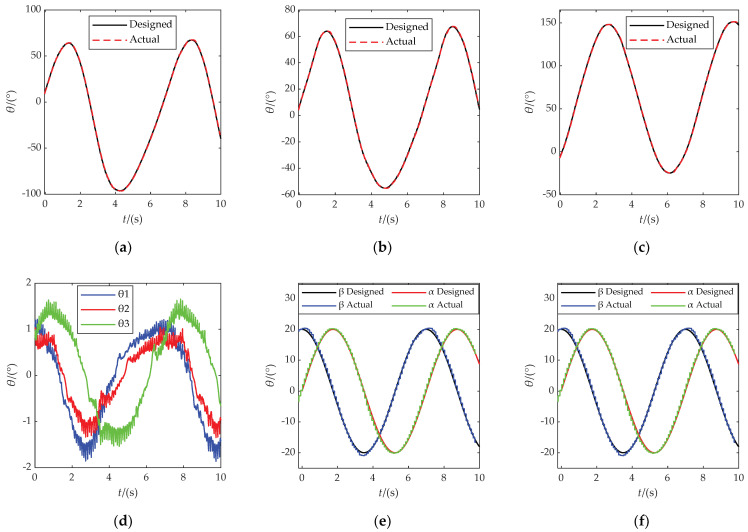
(**a**) The motion law of motor 1; (**b**) the motion law of motor 2; (**c**) the motion law of motor 3; (**d**) the following error of motor; (**e**) the rotation law of the rotating center; (**f**) the following error of the rotating center.

**Table 1 micromachines-13-00950-t001:** Motion range of the ankle joint.

Dorsiflexion: 0°–30°	Plantar flexion: 0°–40°
Adduction: 0°–30°	Abduction: 0°–20°
Varus: 0°–20°	Valgus: 0°–15°

## Data Availability

The original data contributions presented in the study are included in the article; further inquiries can be directed to the corresponding authors.

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
