# Peer review of "Design and Experimental Research of 3-RRS Parallel Ankle Rehabilitation Robot"

_micromachines, 2022, doi:10.3390/mi13060950_

Round 1

Reviewer 1 Report

 A 3-RRS parallel ankle rehabilitation robot (PARR) was proposed. The robot can realize the single and compound ankle rehabilitation training.

1. The motion of ankle joint is approximated as the spherical joint. This coclusion has been verified. So, Sec. 2.1. Ankle Motion Mechanism should be delelted.

2. Auhtor should compare with spherical 3RRR parallel mechanisms in following articals, and explian meritr  of proposed 3RRS parallel mechanism.

Design and kinematic analysis of a 3-RRR spherical parallel manipulator reconfigured with four–bar linkages
Wu, Guanglei (School of Mechanical Engineering, Dalian University of Technology, Dalian; 116024, China); Bai, Shaoping Source: Robotics and Computer-Integrated Manufacturing, v 56, p 55-65, April 2019

 spiral theory is replaced by screw theory.

kinematics analysis only include displacment. Works id weak. The velocity and acceleration models should be established.

Author Response

Dear Reviewers, The relevant questions have been answered and are attached.

Reviewer 2 Report

The work deals with a relevant scientific problem, but has some major flaws:

0. The research itself does not seem to fit the topic of the special issue. The subject of the paper is not microrobots, and not wearable robots either, although the authors mention that the latter are used for ankle rehabilitation. Nowhere in the paper authors present their reason why this paper should belong to this particular Special Issue. However, the decision is up to the editor and not reviewer.

1. Review (lines 40-57) seems to be insufficient. Only 4 robots are mentioned with 6 references. For instance, in one of the recent reviews of only parallel architectures in ankle rehabilitation 16 different mechanism are mentioned

https://doi.org/10.1186/s12984-021-00845-z

2. Paragraph 3.2 is badly written. First, the spiral theory appears to be the screw theory. All the used terms are unfamiliar. Moreover, reference [23], that is cited in the context, has word "screw" and not "spiral" in its title, see https://doi.org/10.1155/2021/6694621
Assuming that the "kinematic spiral" is a twist, the spherical joint will have three twists associated, not one as it is implied in (1). In (2) all rows are designated to the same symbol. The symbol in the used in (3). This is simply incomperhensible. In any case, there will be at least two non-zero moment parts of motors (twist) associated with spherical joint. One axis of the joint can be selected to pass through point O, but two others will not, hence the non-zero moment part of the motors. Equation (4) is written in a vague form without any clarification.
Overall, I suggest to completely rewrite 3.2 from scratch with commonly accepted terminology of the screw theory.
3. Line 276. Error of 2.5 mm for simulation is huge. It is a good result for a physical experiment but not for a model with ideal links and joints. This fact indicates that tere is a possibility of the mathematical error in the theoretical models.

There are also several minor issues:

1. Line 62. The claim that the mechanism has "unique properties" is not supported.

2. Lines 145-147. The Z-axis claimed to be perpendicular and parallel to the static platform at the same time.

3. Lots of typos.

Assuming that the editors decide that the topic of this paper is valid for the SI, I suggest major revision. The main focus should be to rewrite 3.2 and make theoretical part more clear to the reader. The paper must be thoroughly edited.

Author Response

(The authors gave the same response as above.)

Reviewer 3 Report

Abstract

Line 12-22. Please, briefly include some numerical results. An abstract must also summarize the article's main findings and indicate the main conclusion.

Introduction

Lines 35-36. The authors say: “...the number of doctors specialized in rehabilitation training is seriously insufficient.” Please, add a reference to support the mentioned statement.

56-57. The authors say: “…However, the configuration design and human machine compatibility of ankle rehabilitation robot still need to be improved”. What are the issues that still need to be improved?  What issues do current design configurations have? What human-machine compatibility issues do current designs have? and which of all the issues your design solves? Please, include in the introduction the information that answers these questions.

Mechanical Structure

Please, improve quality and size of Figure 2 and Figure 3.

Robot Kinematics Analysis

Please, increase the size of Figure 4 and Figure 5.

Line 180. Where are the angles α, β, γ of the moving platform in the Kinematics model of 3-RRS parallel mechanism in Figure 4?  All the geometric variables used in the kinematic equations must be indicated in the used kinematic schemes.

Lines 245-246. The authors say: “Compared with the motion range of the ankle given in Table 1, it can be found that the rehabilitation training of dorsiflexion /plantar flexion can be completed around 60% to 75%, and the other motions can be completed at 100%. It was shown that the PARR meets the actual rehabilitation training requirements”. Why the authors claim that achieving 60% to 75% of the motion range of dorsiflexion/plantar flexion is enough to meet the actual rehabilitation training requirements? Does not ankle rehabilitation require to complete 100% of the range of motion of dorsiflexion/plantar flexion?

Figure 6. The series labels must not cover the plotted trajectories. 

Multibody Simulation Analysis

Figure 10. The series labels must not cover the plotted trajectories. 

Lines 261-263. Dear authors, why the simulation model of the PARR in Figure 8 appears to not have the same number of joints as the original model in Figure 2? It seems that the three links (each one named Crank Shaft according to Figure 3) connected to the base revolute joints in Figure 2 have become a single rigid body in the model of Figure 8.

Lines 261-263. Does the simulation model respect the restrictions of the type of joints required? That is, for each branch chain, the two revolute joints and the spherical joint were really considered in the simulation model? Or in this case you only program the kinematic equations but Simulink has difficulties in reproducing joint and link behavior of the branch chains?

Lines 250-263. I am afraid that this simulation model is not really reproducing the behavior of the mechanism, if I’m wrong, then more details should be provided in the Simulation Model subsection so that readers who use other types of simulation software (for example SolidWorks, Inventor, Catia) can understand why your simulation model in Figure 8 does not seem to reproduce your proposed mechanism in Figure 2.

Lines 250-260. Please, describe in more detail the modules from figure 7. Furthermore, when you describe the modules of the simulation from figure 7, please indicate exactly which equations you programmed there by indicating the number of the equation according to the number you gave them in the kinematic model section. If you did no use exactly the equations from your kinematic model due to some kind of simplification then you should write the equation used.

Complementary simulations (dynamic analysis and FEM analysis) are needed to characterize the mechanism performance in terms of torques, velocities, accelerations, stiffness, stress, factor of security.

Prototype Experiment

Line 286. For implementing the “Position speed double closed-loop control strategy” do you use two PID controls? Please, clarify this in the text.

Figure 14. The series labels must not cover the plotted trajectories. 

293-301. The authors must specify what strategy they used to carry out the measurement of the trajectory reached by the motors (Figure 14a-c) and the “Rotation law of rotating center” in Figure 14e.

293-301.The methods for testing and the experiment layout must be described.

293-301. The trajectory performed by the end-effector must be measured. Measuring the trajectory of the motors is not enough to characterize the mechanism performance regarding the task, since the trajectory that will have a direct effect on the patient will be the end-effector trajectory.

Figure 13. To carry out a prototype experiment using images, the camera must be in a fixed position with respect to the mechanism, it seems, in Figure 13, that the camera has a different position every time. Was the image processing method used to measure the center of rotation? If so, the design of the experiment appears to have deficiencies since there is no a fixed camera position.

Line 301. The authors say: “The reliability and safety of the ankle rehabilitation robot was verified”. However, the experiments that were carried out by the authors are not enough to support this claim. To say that the robot is safe, additional experiments are needed. Also, to say that the robot is reliable, additional experiments are needed.

280-301. In general, prototype experiment section must be improved describing the used methods and the experiment layout. Furthermore, the lab prototype used for the experimentation must be described. Additionally, the discussion of the results must be deeper and enriching. It is needed to say, from the experiment results, what the prototype needs to be improved or what are the future plans. In practice, did your mechanism show more advantages than the existing ones? which of all the issues your design solved? 

Please remember, the instructions for authors “Authors should discuss the results and how they can be interpreted in perspective of previous studies and of the working hypotheses. The findings and their implications should be discussed in the broadest context possible and limitations of the work highlighted. Future research directions may also be mentioned. This section may be combined with Results.”

Conclusions

305-326. It seems that the conclusions only repeat what the paper contains. Please, add some numerical results to support the conclusions.

Line 336. The “Data Availability Statement” is empty.

Author Response

(The authors gave the same response as above.)

Round 2

Reviewer 1 Report

kinematics analysis only include displacment. Works is weak. The velocity and

acceleration models should be established. Force analysis should be conducted.

Author Response

Dear reviewers, the responses to the relevant questions are attached. Please take note.

Reviewer 2 Report

Unfortunately, I find that 3.2 was not editet properly. Almost all the flaws are still there. All rows in (2) are denoted by the same symbol "$" and therefore must be equal according to this notation. Spherical joint twsist still have zero moment parts, which is just simply worng. And again all equals "$" in (3) and (4).
In the answer on point 3 authors write that there was an error in the software, but it is not mentioned in the paper, and 2.5 mm error assumed to be acceptable for a simulation.
Overall, I must conclude that the authors failed to address the mentioned issues almost completely and therefore I suggest to reject the paper.

Author Response

(The authors gave the same response as above.)

Reviewer 3 Report

The authors have improved the paper.

Author Response

(The authors gave the same response as above.)

Round 3

Reviewer 2 Report

I find that authors have edited the questioned issue. I have no more objections.